# A Generative Augmentation Framework for Contrastive Learning

## Abstract

Contrastive learning has achieved unprecedented levels of accuracy on computer vision applications in recent years. However, in this time, the image augmentations used in these frameworks have remained, for the most part, unchanged. We propose a new augmentation strategy, GenCL, a Generative Augmentation Framework for Contrastive Learning, which utilizes generative modeling to augment images for forming positive pairs. Unlike geometry and color augmentations, GenCL is able to change high-level visual features in images, such as the background, positioning, and color schemas of objects. Our results show that adding these generative augmentations to the suite of augmentations typically used in contrastive learning significantly increases downstream accuracy. In our work, we (1) outline the neural network architecture used in GenCL, (2) use ablation studies to optimize the hyperparameters used in our generative augmentations, and (3) provide a cost-benefit analysis of our implementation in a contrastive learning setting. With these findings, we show that leveraging generative models can significantly increase the performance of contrastive learning on self-supervised learning benchmarks, providing a new avenue for future contrastive learning research.

## 1 Introduction

Extracting representations from images using self-supervised learning has produced state-of-the-art (SOTA) accuracy on the ImageNet dataset, even outperforming supervised learning methods (Tomasev et al., 2022). Contrastive learning (CL) frameworks, in particular, can produce highly accurate visual representations that can be applied to a wide range of applications in computer vision. However, much of the research that goes into improving CL is the development of better frameworks and encoder models (Chen et al., 2020a;b; He et al., 2020; Chen et al., 2020c; 2021; Mitrovic et al., 2020; Tomasev et al., 2022; Grill et al., 2020; Lee et al., 2021; Zbontar et al., 2021; Bardes et al., 2021; 2022). In comparison, little research has been done on developing new augmentations for the forming of positive pairs. Previous CL research typically uses a standardized set of augmentations, which has remained largely unchanged over the years. This group of augmentations is comprised of cropping, flipping, color jitter, Gaussian blur, greyscale, and solarization. These augmentations can be grouped into two types: geometry and color augmentations. Geometry augmentations affect the positioning of pixels, while color augmentations affect the pixels' values. Geometry and color augmentations are easy to implement and have fast, near-instantaneous inference times.

Instead of geometry and color augmentations, our work uses generative augmentations, which we study in a self-supervised learning setting. Generative augmentations utilize modern image editing models such as Generative Adversarial Networks (GANs), Image Transformers (ITs), and Diffusion Models (DMs) to augment images. Unlike geometry and color augmentations, which only augment images at the pixel level, generative augmentations are able to augment high-level visual features (Figure 1). These high-level features include visual elements such as the background, viewing angle, and color schema of an image. The range of image editing techniques made possible by modern generative models is extensive, as is the number of generative models themselves (Xia et al., 2022; Khan et al., 2022; Yang et al., 2022). For this reason, we propose two generative augmentation techniques, noise-based and mask-based augmentations, which we believe can be implemented across a wide range of generative models.

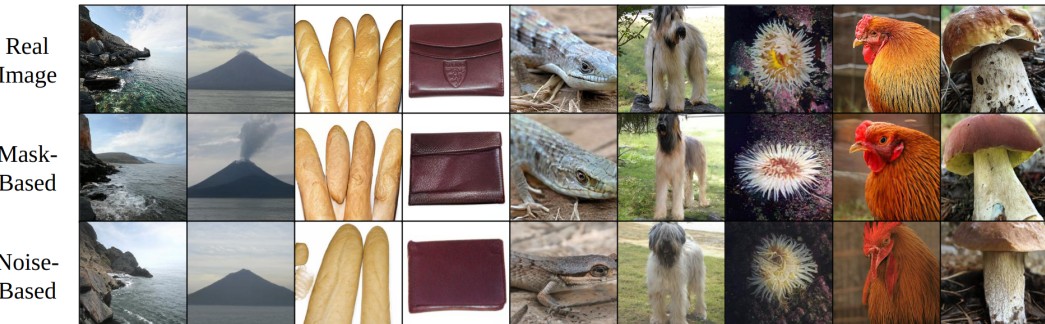

Figure 1: Samples from our mask-based and noise-based augmentations on the ImageNet dataset.

In this paper, we first define both CL and modern image editing models and how they relate to GenCL in Section 2. We then outline our neural network architecture and how it applies image editing models for the forming of positive pairs. The two variations of our architecture, noise-based and mask-based augmentations, are discussed in detail in Section 3. In the following Section 4, we analyze the results of GenCL using standard self-supervised learning benchmarks, as well as several other performance benchmarks. A related works section is provided in Section 5, which describes previous generative augmentation strategies applied to the ImageNet dataset. Finally, a summary of our results and a brief conclusion of GenCL can be found in Section 6, followed by a future works section in Section 7.

## 2 BACKGROUND

GenCL leverages the image editing capabilities of generative models to augment images for the forming of positive pairs in CL. We first briefly outline the basic structure of CL frameworks, followed by a high-level overview of editing images with generative models.

### 2.1 CONTRASTIVE LEARNING

CL frameworks can be broken down into four parts: the augmenter, encoder, projector, and loss function. An image is inputted into the augmenter, which applies a set of randomly sampled augmentations to the image. These augmentations are applied twice to a single image, creating a distinct pair of images known as a positive pair. Due to the random nature in which the augmentations are applied, the two images in the positive pair will have varying visual features. The positive pair is then passed into the encoder, which reduces each 2D image down into a 1D visual representation. Once the two images from the positive pair are encoded into these visual representations, they are each inputted into the projector, a shallow neural network, which outputs a 1D encoding. This pair of outputted encodings is then used for calculating the CL loss. The resulting loss is then backpropagated through both the encoder and projector. Once training is completed, the projector is typically discarded, and the encoder can then be used to encode images down into visual representations (Chen et al., 2020b). These representations can then be applied to a wide range of computer vision applications such as object detection and semantic segmentation (Bardes et al., 2021; 2022). This is done by first inputting the images into the encoder, which outputs their corresponding visual representations. These representations are then inputted into another model, typically a single-layer neural network, which outputs the predictions for the selected downstream task. Current CL methods offer several advantages over supervised learning methods, i.e., downstream accuracy, task flexibility, and semi-supervised applications.

### 2.2 IMAGE EDITING MODELS

In computer vision, generative models are deep neural networks that are able to synthesize artificial images, commonly known as deep fakes. The top-performing generative models all use either one or some combination of the GAN, IT, or DM architectures in their design, as these architectures

produce the most photo-realistic images (Gao et al., 2023; Kim et al., 2023; Peebles & Xie, 2023; Hang et al., 2023; Sauer et al., 2022; Kang et al., 2023). In addition to generating completely synthetic images, generative models also have the ability to add artificial visual features to real images. To edit an image, a generative model must first map the image into its underlying feature map. This is typically done using either a feature extraction model or an image inversion method. A feature extraction model can map an image directly into a generative model's feature map, which outputs an image with similar visual features. Alternatively, image inversion methods first initialize a random feature map for the generative model, which outputs a synthetic image. The difference between the inputted image and the synthetic image is then minimized as the values of the feature map are backpropagated for a number of iterations. Using generative models with feature extractors already built into their neural network architecture is more apt for editing large sets of images, as the image inversion process is highly iterative and computationally expensive. Once the inputted image is mapped into a generative model's feature map, the feature map can be manipulated in a number of ways that alter the visual features of the outputted images. The synthetic visual features added by modern image editing models are often of such high quality that they are difficult to discern from real visual features.

## 3 METHODOLOGY

In our study of generative augmentations, we use two different approaches for manipulating a generative model's feature map. Both approaches are efficient at augmenting large numbers of images, can scale to high resolutions, and can be applied to a large array of generative models.

### 3.1 NOISE-BASED AUGMENTATIONS

The first of our generative augmentation approaches is noise-based augmentations, outlined in Figure 2. In this approach, we map an image into a generative model's feature map and then insert noise into the feature map. For generative models that use a GAN architecture, noise can be added through the inputted noise vector. For IT and DM architectures, noise can be added in a number of ways, such as dropout layers or adding small amounts of Gaussian noise directly to the feature map's values. The noisy feature map is then fed into the generative model, which outputs an image that resembles the original image but with some variations in its visual features. The inputted image and outputted images can then be used to form a positive pair.

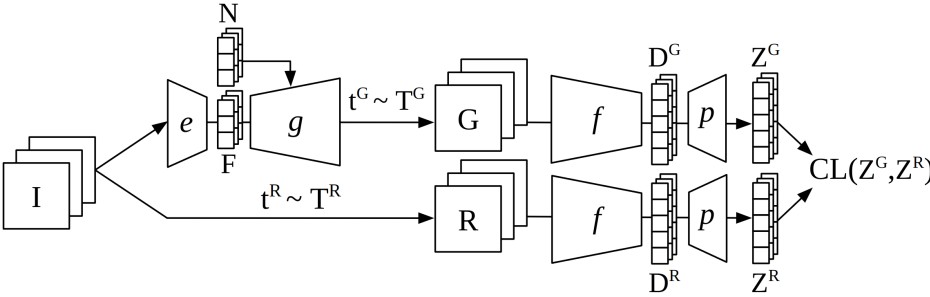

Figure 2: NOISE-BASED AUGMENTATION architecture within a CL framework. Given a batch of images $\mathbf{I}$, two views are produced. The first view is fed into the feature extractor $e$, which outputs the corresponding feature maps $\mathbf{F}$. Noise vectors $\mathbf{N}$ are then sampled, paired with the feature maps $\mathbf{F}$, and fed into the generator model $e$, which outputs the generatively augmented images. A subset of augmentations $\mathbf{t}^G$ is then sampled from a set of geometry and color augmentations $\mathbf{T}^G$ and applied to the generatively augmented images, creating a batch of images $\mathbf{G}$. A second subset of augmentations $\mathbf{t}^R$ is then sampled from a set of geometry and color augmentations $\mathbf{T}^R$ and applied to the second view, creating a batch of images $\mathbf{R}$. The positive pair $(\mathbf{G}, \mathbf{R})$ is then fed into the encoder model $f$ creating the visual representations $(\mathbf{D}^G, \mathbf{D}^R)$. The representations are then fed into the projector model $p$, producing the encodings $(\mathbf{Z}^G, \mathbf{Z}^R)$, which are inputted into the $\mathbf{CL}$ loss.

For our implementation of noise-based augmentations, we use a pre-trained Instance-Conditioned Generative Adversarial Network (ICGAN) as our generative model. The ICGAN is well suited to generative augmentations. Its architecture includes a feature extraction model, allowing for fast, non-iterative inference times when mapping an image to a feature map. The ICGAN also scales to 256x256 image resolution on the ImageNet dataset in a completely unsupervised manner Casanova et al. (2021). CL applications often exist in a semi-supervised learning setting, where only a small number of the training images are labeled, making unsupervised generative models, like the ICGAN, ideal for these applications. Unfortunately, unsupervised generative models produce images with significantly poorer image quality on large-scale applications like ImageNet compared to supervised generative models Casanova et al. (2021); Donahue & Simonyan (2019). When measuring the image quality of synthetic images produced by generative models, the Fréchet Inception Distance (FID) is the standard metric (Heusel et al., 2017). To measure FID, we first encode the real and synthetic images using an InceptionV3 model and then compare the encodings' distributions. The ICGAN suffers from a relatively low FID score, achieving an FID of 15.6 on the ImageNet dataset (Casanova et al., 2021). In comparison, other supervised GAN variants, such as the StyleGAN-XL, are able to achieve an FID as low as 2.3 on the ImageNet dataset (Sauer et al., 2022).

## 3.2 MASK-BASED AUGMENTATIONS

Our second generative augmentation strategy is a mask-based approach, detailed in Figure 3. Similar to our noise-based approach, we first map an image into our generative model's feature map using a feature extraction model. We then mask values from the feature map and use an in-painting model to predict the missing values. The difference between the original feature map and the predicted feature map will increase as the number of masked values increases. This is due to the decreasing amount of information being fed to the in-painting model. We also found that increasing the masked values' adjacency increases the difference between the original and predicted feature maps. Once the masked values are replaced with the predicted values, the feature map is inputted into the generative model, which outputs the generatively augmented image. The difference in visual features between the inputted and outputted images is directly related to the difference between the original and predicted feature maps. Therefore, we can increase the difference in visual features between the inputted and augmented images by increasing the size and adjacency of our masked values. Unlike noise-based augmentations, mask-based augmentations can be applied to feature maps that follow a discrete distribution as well as a continuous distribution. Vector quantization methods are particularly well-suited to mask-based augmentations as they use a discrete distribution and are commonly used in generative modeling (Chang et al., 2022; Esser et al., 2021; Razavi et al., 2019). Unlike noise-based augmentations, mask-based augmentations require an in-painting model to manipulate the values of the feature map.

We use a pre-trained Masked Generative Image Transformer (MaskGIT) as the generative model in our mask-based approach (Chang et al., 2022). The MaskGIT model uses a Vector Quantized Generative Adversarial Network (VQGAN) to encode 256x256 resolution images into 16x16 feature maps with a codebook of 1,024 discrete values (Esser et al., 2021). Our masking strategy masks every discrete value except the ones on the outer edges of the 2D feature map. We found this masking strategy maintains a higher structural similarity between the original and augmented image compared to our noise-based approach while also providing a good amount of variation in the visual features. The in-painting model used to predict the masked values of the feature map is a bidirectional IT. Because we use a bidirectional IT instead of a unidirectional IT, the masked values can be predicted in parallel. The predicted values with the highest confidence are kept and combined with the input, while the remaining values are discarded. This is repeated for 12 iterations until no masked values remain. This feature map is then inputted into the VQGAN's generator model, which outputs the augmented image. The MaskGIT's use of the VQGAN and bidirectional IT architectures results in significantly fewer iterations when augmenting images than other ITs or DMs (Chang et al., 2022). Bidirectional ITs are also well suited to mask-based augmentations due to their innate in-painting and out-painting capabilities. MaskGIT provides a much higher image quality than ICGAN, achieving an FID of 4.02 on the ImageNet dataset (Chang et al., 2022). Like the ICGAN, the MaskGIT includes a feature extraction model in its architecture; however, unlike the ICGAN, the MaskGIT is trained in a supervised manner. Because supervised generative models trained on large image datasets like ImageNet have excellent performance when editing out-of-domain images, we

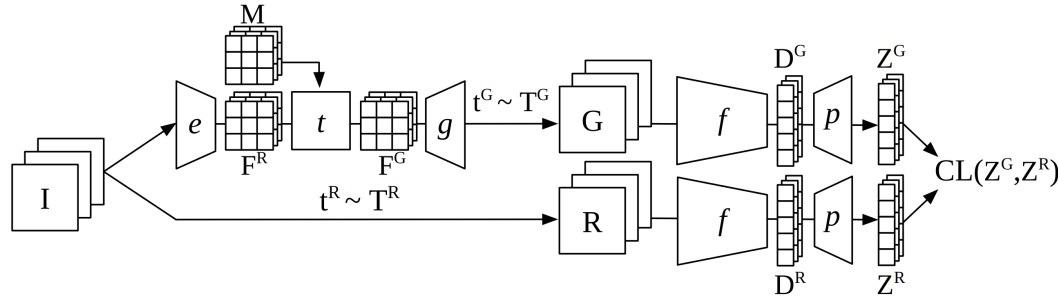

Figure 3: MASK-BASED AUGMENTATION architecture within a CL framework. Given a batch of images $\mathbf{I}$, two views are produced. The first view is fed into the feature extractor $e$, which outputs the corresponding feature maps $\mathbf{F}^R$. Mask vectors $\mathbf{M}$ are then sampled and applied to the feature maps $\mathbf{F}^R$. The masked feature maps are fed into the in-painting model $t$, which outputs the predicted feature maps $\mathbf{F}^G$. $\mathbf{F}^G$ is then fed into the generator model $e$, which outputs the generatively augmented images. A subset of augmentations $\mathbf{t}^G$ is then sampled from a set of geometry and color augmentations $\mathbf{T}^G$ and applied to the generatively augmented images, creating a batch of images $\mathbf{G}$. A second subset of augmentations $\mathbf{t}^R$ is then sampled from a set of geometry and color augmentations $\mathbf{T}^R$ and applied to the second view, creating a batch of images $\mathbf{R}$. The positive pair $(\mathbf{G}, \mathbf{R})$ is then fed into the encoder model $f$ creating the visual representations $(\mathbf{D}^G, \mathbf{D}^R)$. The representations are then fed into the projector model $p$, producing the encodings $(\mathbf{Z}^G, \mathbf{Z}^R)$, which are inputted into the $\mathbf{CL}$ loss.

posit they can also successfully be applied to semi-supervised applications (Sauer et al., 2022; Kang et al., 2023; Saharia et al., 2022).

Due to generative models' large computational footprints and slow inference times, we recommend computing generative augmentations before training the CL model for both noise-based and mask-based augmentations. When using the ICGAN and MaskGIT models, we found that generating a single generatively augmented image takes less than a quarter of the time to train a Variance Invariance Covariance Regularization (VICReg) CL model with a ResNet50 encoder for 100 epochs (Bardes et al., 2021). We also found that generating more than one generatively augmented image per training image did not increase downstream accuracy. In addition to the computational costs, storing a single generatively augmented for each training image doubles the dataset's size. However, when the generative augmentations have been pre-computed, the CL model's training speed is virtually unaffected. Similar to geometry and color augmentations, generative augmentations benefit from being applied to a subset of the training images. After a generative augmentation is applied to an image, a set of geometry and color augmentations should also be applied, as this can further increase downstream accuracy.

## 4 RESULTS

When developing our GenCL framework, we found two hyperparameters significantly affect its downstream accuracy. The first hyperparameter is the set of geometry and color augmentations applied to the generatively augmented images, and the second is how often our generative augmentations are applied to images during training. To optimize the first hyperparameter, we apply a pairwise ablation between our generative augmentations and the geometry and color augmentations used in (Bardes et al., 2021). For our second hyperparameter, we apply a sampling rate ablation where we adjust the sampling rate at which our generative augmentations are applied. In our ablation studies, we train several CL models, each using the VICReg CL framework and a ResNet50 as the encoder model (Bardes et al., 2021; He et al., 2015). We train each of these models on the 1,000 class, full-resolution ImageNet dataset using 4 NVIDIA 3090 GPUs. In each of our benchmarks, the encoder models are trained using the same protocol used in (Bardes et al., 2021), with the exception of the batch size being reduced to 800 and the training epochs being reduced to 100 due to hardware restraints. Once accepted, we plan to release our code on GitHub as well as provide instructions

| | Linear | | Semi-Supervised | | | |
|---|---|---|---|---|---|---|
| Augmentations | Top-1 | Top-5 | Top-1 | | Top-5 | |
| | | | 1% | 10% | 1% | 10% |
| Generative | 54.4 | 78.4 | 32.9 | 49.3 | 60.5 | 75.1 |
| Crop + Generative | 62.8 | 84.0 | 40.6 | 58.3 | 68.3 | 82.0 |
| Flip + Generative | 57.4 | 80.5 | 35.7 | 52.6 | 63.8 | 77.8 |
| Solarization + Generative | 55.8 | 79.7 | 32.9 | 50.5 | 61.1 | 76.2 |
| Color Jitter + Generative | 55.7 | 79.3 | 32.5 | 50.1 | 60.5 | 76.0 |
| Gaussian Blur + Generative | 54.2 | 78.3 | 32.1 | 49.3 | 60.0 | 75.2 |
| Greyscale + Generative | 52.7 | 77.3 | 30.6 | 48.4 | 58.5 | 74.6 |

Table 1: PAIRWISE AUGMENTATION ABLATION's Top-1 and Top-5 accuracies obtained with our noise-based augmentations using the VICReg framework and a ResNet50 encoder. The linear classification uses frozen representations on the ImageNet dataset. The semi-supervised classification uses fine-tuned representations on 1% and 10% of the ImageNet dataset.

for downloading the generatively augmented images for both our noise-based and masked-based augmentations.

### 4.1 PAIRWISE AUGMENTATION ABLATION

In our first ablation study, we pair our generative augmentations with each of the six geometry and color augmentations commonly used in CL: cropping, flipping, solarization, color jitter, Gaussian blur, and greyscale. We train a CL model using only our generative augmentations as a baseline. Each model in our pairwise ablation applies generative augmentations to 100% of the training images. The sampling rate and other hyperparameters used for the geometry and color augmentations are identical to those used in (Bardes et al., 2021). For our pairwise augmentation ablation, we only apply our noise-based augmentation strategy, and the results can be seen in Table 3.

Our results show that using generative augmentations with each of the two geometry augmentations (cropping and flipping) significantly boosts downstream accuracy. The cropping augmentation provides the largest increase in downstream accuracy when paired with generative augmentations, increasing the linear top-1 accuracies by 8.4%. Horizontal flipping results in a 3% increase in linear top-1 accuracy, more than double the increase in accuracy from any of the color augmentations. Of the color augmentations, only solarization and color jitter provide any benefit to downstream accuracy when paired with generative augmentations, each increasing the linear top-1 accuracy by ~1%. Pairing our generative augmentations with either the Gaussian blur or greyscale augmentations resulted in a lower accuracy than our baseline model. Our results show that our generative augmentations benefit significantly when paired with geometry augmentations but also benefit when paired with the solarization or color jitter augmentations.

### 4.2 SAMPLING RATE ABLATION

Similar to geometry and color augmentations, we apply generative augmentations to a subset of the training images. We randomly sample images with a sampling rate of $\gamma$. For the selected images, we apply the generative augmentation, then apply the cropping, flipping, solarization, and color jitter augmentations, as these increased the linear top-1 accuracy in our pairwise ablation. If an image is not sampled, no generative augmentation is applied, but the cropping, flipping, solarization, color jitter, Gaussian blur, and greyscale augmentations are applied. For our baseline model, we set $\gamma$ to 0 so that no generative augmentations are applied during training.

In the results for our noise-based sampling rate ablation setting $\gamma$ to 0.25 resulted in the highest accuracy in both the linear and semi-supervised evaluations (Table 2). The next best accuracies were when $\gamma$ was set to 0.5, 0 (baseline), 0.75, and 1.0, respectively. When using a $\gamma$ higher than 0.5, we see the downstream accuracy dip below our baseline, showing that our noise-based augmentations degrade downstream accuracy when applied to more than half the training images. Although our best-performing noise-based augmentation increases the linear top-1 accuracy, it only does so by

|  | Linear | | Semi-Supervised | | | |
|---|---|---|---|---|---|---|
| $\gamma$ | Top-1 | Top-5 | Top-1 | | Top-5 | |
| | | | 1% | 10% | 1% | 10% |
| 0% | 65.8 | 86.6 | 42.9 | 60.7 | 70.2 | 84.1 |
| 25% | 66.3 | 87.0 | 43.9 | 61.1 | 71.4 | 84.3 |
| 50% | 65.9 | 86.7 | 42.9 | 60.5 | 70.4 | 83.6 |
| 75% | 65.3 | 86.1 | 41.7 | 59.6 | 69.5 | 82.9 |
| 100% | 64.0 | 85.0 | 40.7 | 58.9 | 68.3 | 82.5 |

Table 2: SAMPLING RATE ABLATION's Top-1 and Top-5 accuracies obtained with our noise-based augmentations using the VICReg framework and a ResNet50 encoder. The linear classification uses frozen representations on the ImageNet dataset. The semi-supervised classification uses fine-tuned representations on 1% and 10% of the ImageNet dataset.

0.5%. This low increase in accuracy is likely due to the low image quality of the ICGAN model, which is evidenced by the results from our mask-based approach's sampling rate ablation.

For our mask-based sampling rate ablation, the highest downstream accuracy was achieved when $\gamma$ was set to either 0.5 or 0.75, which had approximately the same accuracy in both the linear and semi-supervised evaluations. The next highest accuracies were when $\gamma$ was set to 0.25, 1, and 0 (baseline), respectively. This differs from our noise-based augmentations in that any amount of mask-based augmentations increases our downstream accuracy over the baseline. Our best-performing linear top-1 accuracy was achieved when $\gamma$ was set to 0.5, achieving an accuracy of 68.5%, which is 2.7% higher than our baseline. For context, the difference in linear top-1 accuracy using a ResNet50 encoder between the SOTA ReLICv2 CL framework and the C-BYOL CL framework is 1.5% (Tomasev et al., 2022; Lee et al., 2021). Because of this significant increase in accuracy and the fact that adding any amount of mask-based augmentations increases accuracy, we believe this shows a clear benefit to using generative augmentations in a CL setting.

### 4.3 COST-BENEFIT ANALYSIS

Applying generative augmentations incurs a significantly greater computational cost compared to geometry and color augmentations. Modern image editing models require substantial computational resources and are often highly iterative, making large-scale applications difficult. However, we posit certain generative models, like the ICGAN and MaskGIT, are particularly well-suited to augmenting large numbers of images. To show this, we profiled the performance of our noise-based (ICGAN) and mask-based (MaskGIT) augmentations as well as the cropping, color jitter, and Gaussian blur augmentations by measuring each augmentation's linear top-1 accuracy, FID, Structural Similarity Index (SSIM), inference time, and the number of parameters used (Table 4). For the noise-based and mask-based augmentations' linear top-1 accuracy, we trained a model using the VICReg framework and a ResNet50 encoder on the ImageNet dataset. To calculate the linear top-1 accuracy of the crop, color jitter, and Gaussian blur augmentations, we take the results from (Chen et al., 2020a) and add the difference in linear top-1 accuracy between the VICReg and SimCLR models presented in (Bardes et al., 2021), which are indicated by †. For each image in ImageNet's training dataset, we calculate the FID and SSIM between the original and augmented image and aggregate the results. To calculate inference time, we average the time taken to augment 1,000 images from the ImageNet dataset using a batch size of one. We also measure the number of parameters the ICGAN and MaskGIT models use when augmenting an image.

The augmented images produced by the MaskGIT model have the highest linear top-1 accuracy, outperforming the ICGAN's augmented images by more than 12% and each of the geometry and color augmented images by at least 32%. This large difference in accuracy between the MaskGIT and ICGAN augmentations is likely due to the difference in both FID and SSIM. The FID of the images augmented by the MaskGIT is also better than those augmented by the Gaussian blur augmentation but worse than those augmented by the cropping and color jitter augmentations. The SSIM of the MaskGIT's augmented images is 50% higher than that of the ICGAN's. The MaskGIT's inference time is 5 times slower than the ICGANs and 27-171 times slower than the geometry and color augmentation's. The number of parameters used by the MaskGIT model is twice as many as the ICGAN

| $\gamma$ | Linear | | Semi-Supervised | | | |
|---|---|---|---|---|---|---|
| | Top-1 | Top-5 | Top-1 | | Top-5 | |
| | | | 1% | 10% | 1% | 10% |
| 0% | 65.8 | 86.6 | 42.9 | 60.7 | 70.2 | 84.1 |
| 25% | 67.9 | 87.9 | 47.2 | 62.8 | 73.7 | 85.3 |
| 50% | 68.5 | 88.3 | 48.3 | 63.3 | 74.7 | 85.5 |
| 75% | 68.5 | 88.4 | 48.5 | 63.2 | 74.6 | 85.5 |
| 100% | 67.5 | 87.5 | 48.5 | 62.9 | 74.2 | 85.1 |

Table 3: SAMPLING RATE ABLATION's Top-1 and Top-5 accuracies obtained with our mask-based augmentations using the VICReg framework and a ResNet50 encoder. The linear classification uses frozen representations on the ImageNet dataset. The semi-supervised classification uses fine-tuned representations on 1% and 10% of the ImageNet dataset.

| | Linear Top-1 | FID | SSIM | Inference Time (ms) | Parameters |
|---|---|---|---|---|---|
| Noise-Based | 54.4 | 21.13 | 0.21 | 19.82 | 116M |
| Mask-Based | 67.8 | 3.70 | 0.32 | 98.93 | 227M |
| Crop | 35.0[†] | 2.48 | 0.27 | 0.58 | N/A |
| Color Jitter | 22.7[†] | 0.23 | 0.87 | 3.70 | N/A |
| Gaussian Blur | 6.5[†] | 3.72 | 0.88 | 0.64 | N/A |

Table 4: COST-BENEFIT ANALYSIS's linear Top-1 accuracy, FID, SSIM, inference time, and number of parameters for the noise-based (ICGAN), mask-based (MaskGIT), crop, color jitter, and gaussian blur augmentations on the ImageNet dataset.

model. Augmenting all of the 1,281,167 training images in the ImageNet dataset using a batch size of one would take approximately 35 hours with the MaskGIT model and approximately 7 hours with the ICGAN. In comparison, training a VICReg CL model with geometry and color augmentations for 100 epochs with a batch size of one would take approximately 1,790 hours, which is 51 times longer than augmenting all the images with the MaskGIT model and 255 times longer than augmenting all the images with the ICGAN model. Increasing the batch size would decrease each of these times considerably.

## 5 RELATED WORK

Several projects have employed generative models for the forming of positive pairs. Of these, only the GenRep (Jahanian et al., 2022) and COP-Gen (Li et al., 2022) generative augmentation strategies have been applied to the ImageNet dataset. Both of these generative augmentation strategies form their positive pairs with two completely synthetic images. This differs from our GenCL implementation, which forms the positive pair with one real image, which is then generatively augmented to produce the second image in the positive pair. GenRep and COP-Gen's implementations require no feature extraction model, as the first image's feature map is initialized randomly and then manipulated to create the second image in the positive pair. The downside to the GenRep and COP-Gen generative augmentation strategies is that they cannot be applied to real images, as they lack a way to map images to their generative model's feature map. Using only synthetic images likely causes significant information loss in the CL model, resulting in poor downstream accuracy. This is evidenced by their highest linear top-1 accuracy on the ImageNet dataset being 53.3%, 15.2% less than our highest-performing GenCL model. Our benchmarks and those provided in the GenRep and COP-Gen implementations use the same ResNet50 encoder model and are trained for the same number of epochs. The GenRep and COP-Gen implementations are identical except in how they manipulate their generative model's feature map. GenRep manipulates their generative model's feature map in a method known as steering, which causes the changes between the two synthetic images to result in visual changes that mimic the effects of changes in luminance, zoom level, and shifts along the $xy$ axes (Jahanian et al., 2020). COP-Gen manipulates their generative model's feature map using a two-layer neural network. This shallow neural network is first trained using the negative value of the Information Noise-Contrastive Estimation (InfoNCE) as its loss function.

## 6 CONCLUSION

Our work presents GenCL, a framework for augmenting images with modern image editing models in a CL setting. We designed two approaches for GenCL, one noise-based and the other masked-based. We found that both approaches increase downstream accuracy but that our mask-based approach outperformed our noise-based approach considerably. This is likely due to the higher FID scores achieved by our masked-based approach's augmented images. Synthetic images with a lower FID score contain more accurate and detailed visual features (Heusel et al., 2017). This higher-quality visual information can then be transferred to the CL model, resulting in improved downstream accuracy. While developing GenCL, we found that applying generative augmentations that change too many visual features can lower downstream accuracy. For this reason, a higher structural similarity between the original and generatively augmented image also likely increases downstream accuracy. Our results show that using geometry augmentations alongside GenCL significantly improves downstream accuracy. However, the increase in accuracy from pairing color augmentations with our generative augmentations is considerably less. This is likely because the generative augmentations produced by GenCL already alter the color schemas in images, making additional color augmentations redundant. Similar to geometry and color augmentations, we found that applying GenCL to a subset of the training images also results in better downstream accuracy. Finally, we analyze the costs of implementing our generative augmentations in a CL setting. Modern image editing models often have slow, iterative inference times. However, certain generative models, like the ICGAN and MaskGIT, have much faster inference times than other generative models when augmenting images, resulting in lower computational overhead when applied to large-scale applications. Because of the slower inference time and large computational footprint of generative models, generative augmentations should be calculated beforehand, as calculating them while training your CL model will considerably increase GPU/TPU memory consumption and training time. The image editing capabilities of modern generative models are truly remarkable and have had profound effects across the realm of computer vision. Our results illustrate that our GenCL framework adeptly utilizes these generative models to augment images for the forming of positive pairs, resulting in improved downstream accuracy and efficient computational scalability for CL models in a self-supervised setting.

## 7 FUTURE WORK

With our GenCL framework, we utilize both noise-based and mask-based augmentations, but many other image editing methods exist in current generative models that could also potentially be used for generative augmentations. In particular, generative models like MaskGIT, which have strong in-painting and out-painting capabilities, could be utilized in a number of ways. For example, reducing an image's resolution and then extrapolating the pixels around the image with MaskGIT could create a generative augmentation that produces a reverse cropping effect. 3D-aware vision models that utilize neural radiance fields are able to shift the viewing position and angle of an image in a 3D space, providing another possible avenue for future generative augmentation research (Sargent et al., 2023; Skorokhodov et al., 2023; Cai et al., 2022). Recently, generative models that convert text-to-image and image-to-text have received much attention (Ramesh et al., 2022; Yu et al., 2022; Kang et al., 2023). By mapping an image to text and then mapping that text to a synthetic image is yet another example of the many modern image editing methods that could potentially be used for generative augmentations.

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
