# OpenReview forum: "A Generative Augmentation Framework for Contrastive Learning"
_ICLR.cc/2024/Conference — ICLR 2024 Conference Withdrawn Submission_

### Official Review · Reviewer_7GZG · 2023-10-18

**Soundness:** 2 fair
**Presentation:** 4 excellent
**Contribution:** 2 fair
**Rating:** 3
**Confidence:** 4

**Summary:**

The authors proposed GenCL, which combines modern generative models with contrastive learning. Specifically, GenCL uses generative models to create better-augmented images when compared with heuristic augmentations for contrastive learning. The empirical results show that GenCL improves the downstream classification performance on ImageNet.

**Strengths:**

- The experimental results seem promising, as shown in Table 1.
- The paper is well-written and easy to follow.

**Weaknesses:**

(1) The technical novelty of the proposed method is quite limited. The authors used existing well-performing generative models (ICGAN and Mask GIT) as the augmentation module in the standard contrastive learning framework. Using noised representation and in-painting methods to generate augmented images is not new.

(2) The method is only tested on downstream ImageNet classification. Other CL literature evaluates their methods on more CV tasks, like object detection and segmentation.

**Questions:**

- GenCL relies heavily on pre-trained generative models. If the target domain differs from the pre-trained domain, the generative models may fail to produce valid augmented images. This limits the use of GenCL in practice.

- The paper is a good application of generative models and contrastive learning methods. However, it may not be suitable for ICLR due to the lack of novelty.

---

### Official Review · Reviewer_4FGw · 2023-10-28

**Soundness:** 3 good
**Presentation:** 3 good
**Contribution:** 2 fair
**Rating:** 5
**Confidence:** 4

**Summary:**

The paper introduces GenCL, a Generative Augmentation Framework for Contrastive Learning, aiming to improve upon the standard image augmentation methods typically used in contrastive learning. The authors outline the GenCL neural network architecture, detail the two proposed generative augmentation techniques (noise-based and mask-based), and provide a performance analysis of their method on standard self-supervised learning benchmarks.

**Strengths:**

S1 - Generative Augmentations:

Unlike traditional geometric and color augmentations, which predominantly focus on pixel-level modifications, GenCL's generative approach can modify high-level visual features, offering richer and more diverse augmentations.

S2 - Performance Boost:

The results demonstrate that incorporating generative augmentations in contrastive learning can lead to significant improvements in self-supervised learning benchmarks.

S3 - Comprehensive Analysis:

The paper thoroughly evaluates the GenCL approach, detailing the neural network architecture and providing in-depth ablation studies.

**Weaknesses:**

W1 - Effectiveness over contrastive learning framework and downstream tasks:
While the proposed augmentation method is effective, further experiments are expected across standard contrastive learning frameworks such as MoCo, SimCLR, or BYOL[1, 2, 3] to prove the generalization capacity. Meanwhile, I'm also curious whether the augmentation methods boost the performance of tasks like object detection and segmentation.

W2 - Computation cost:
Generative model-based augmentation will unavoidably introduce additional computation. However, how is the computation when compared with native multi-crops methods from InfoMin[4]? If the proposed method needs more computation or the improvements become marginal with multiple crops, then the contribution of this method would be less salient.

W3 - Discussion about related work (minor).
There is a series of works that aims to boost the performance of contrastive learning in the feature space. Either of them aims to generate better positive pairs or provide additional contrast. I'd suggest the author give them sufficient discussion and comparison[5,6,7].

[1] Momentum Contrast for Unsupervised Visual Representation Learning

[2] A Simple Framework for Contrastive Learning of Visual Representations

[3] Bootstrap your own latent: A new approach to self-supervised Learning

[4] What Makes for Good Views for Contrastive Learning?

[5] Towards domain-agnostic contrastive learning

[6] Hallucination Improves the Performance of Unsupervised Visual Representation Learning

[7] Metaug: Contrastive learning via meta feature augmentation

**Questions:**

The main questions are listed in Weaknesses. I'd raise my score if they were appropriately addressed.

---

### Official Review · Reviewer_NnqA · 2023-10-28

**Soundness:** 3 good
**Presentation:** 3 good
**Contribution:** 2 fair
**Rating:** 3
**Confidence:** 4

**Summary:**

The work focuses on utilizing the power of generative models for augmentations (or views) in contrastive learning. The authors propose two methods: noise-based and masked-based augmentations in contrastive learning. The noise-based augmentations perturbs the feature map of the original by adding some Gaussian noise and this noisy feature map is fed into the generative model to synthesize a semantically similar image. The mask-based augmentations masks a portion of the feature map and then uses an inpainting model to recreate the original image. In this work, the authors use MaskGIT for the mask-based augmentations and Instance-Conditioned GAN for the noise-based augmentations. The authors show results on ImageNet-1k dataset with VICReg (a contrastive learning algorithm) where the mask-based augmentations outperforms the baseline  and noise-based augmentations. A comprehensive analysis of the computation cost is also mentioned in the results section.

**Strengths:**

- The concept of replacing hand-crafted augmentations with generative augmentations is intuitive, logical and interesting direction.
- The computation cost benefit analysis is comprehensive. The authors address the biggest problem with generative data augmentation i.e computing data augmentation using a generative model is extremely time and compute-heavy.
- Overall, the paper is well-written. In the related work section, the authors clearly mention the difference between the proposed methods and previous related research: GenRep and COP-Gen.
- Results with Mask-based augmentations with VICReg on ImageNet-1k show better performance than the baseline.

**Weaknesses:**

- The idea of IC-GAN as data augmentation or views in Self Supervised learning has been previously explored in [1]. In Astolfi et. al [1], they integrate IC-GAN with SwAV and show results on ImageNet (Refer Section 5.2 in the paper). The authors should discuss this paper and also compare the results. Particularly, the authors should elaborate on how the proposed noise-based augmentation is different from the one in [1].
- These are my concerns with the results section.
    - The VICReg [2] paper reports a 68.7% Top-1 Accuracy with 100 epochs (Table 4 of VICReg) The authors also run the VICReg for 100 epochs (due to computation constraints). The baseline reported by the authors (0% sampling rate corresponds to the VICReg baseline) in Table 2 and Table 3 is 65.8% Top-1 Accuracy on ImageNet. This is an important point given that the gains with the generative augmentation is less than 2% in most cases. Can the authors clarify the reason behind the same?
    - Previous work [3, 4, 5] has shown that generative data augmentation usually leads to benefit in OOD tasks. The results may be more compelling if the authors can analyse the results on some of the OOD datasets like ImageNet-C, ImageNet-R, ImageNet-Sketch, ImageNet-V2 and ObjectNet to name a few.
    - The proposed noise-based augmentation and mask-based augmentation are not specific to any CL method. From that point of view, the authors should ideally show results with atleast one more CL method apart from VICReg.
    - Some baselines are missing from the tables. Given that GenRep and COP-Gen are closely related, the authors should compare with these methods. In Table 1, the authors should mention the performance of VICReg baseline (It looks like the VICReg baseline outperforms all the generative augmentations in Table 1). Similarly, in Table 3, the authors should also mention MaskGIT baseline performance (Classification Accuracy Score) for reference.
    - Some important ablations missing: Similar to GenRep, the authors can also consider steered latent views as an alternative to Gaussian noise in the noise-based augmentations. Similarly, random masking certain percentage of discrete values can be chosen as an ablation in the mask-based augmentations.

    [1] Astolfi, Pietro, et al. "Instance-Conditioned GAN Data Augmentation for Representation Learning." *arXiv preprint arXiv:2303.09677* (2023) TMLR

    [2] Bardes, Adrien, Jean Ponce, and Yann LeCun. "Vicreg: Variance-invariance-covariance regularization for self-supervised learning." arXiv preprint arXiv:2105.04906 (2021).

    [3] Sariyildiz, Mert Bulent, et al. "Fake it till you make it: Learning transferable representations from synthetic ImageNet clones." CVPR 2023–IEEE/CVF Conference on Computer Vision and Pattern Recognition. 2023.

    [4] Bansal, Hritik, and Aditya Grover. "Leaving reality to imagination: Robust classification via generated datasets." arXiv preprint arXiv:2302.02503 (2023).

    [5] He, Ruifei, et al. "Is synthetic data from generative models ready for image recognition?." arXiv preprint arXiv:2210.07574 (2022).

**Questions:**

1. In Section 5: “This is evidenced by their highest linear top-1 accuracy on the ImageNet dataset being 53.3%, 15.2% less than our highest-performing GenCL model.” Have the authors run GenRep with VICreg? Otherwise, it is not fair to compare the numbers of GenRep and GenCL given that GenRep uses a different generative model and a different contrastive method.
2. The authors mention the below statement in Section 4.3

    "In comparison, training a VICReg CL model with geometry and color augmentations
    for 100 epochs with a batch size of one would take approximately 1,790 hours, which is 51 times longer than augmenting all the images with the MaskGIT model and 255 times longer than augmenting all the images with the ICGAN model."
    I am not sure I understand why the authors are comparing inference time of a generative model with training of VICReg CL model with 1 epoch.

3. The authors can definitely try exploring a combination of mask-based and noise based augmentations to see if they further improve performance.
4. I just wanted to clarify that I do not see any supplementary material. There are a lot of implementation details missing in the paper which are necessary to reproduce the results obtained in the paper. For instance, what is the strength of Gaussian noise used for noise based augmentation? Also additional details about the implementation of mask-based augmentations would make things clearer.

---

### Official Review · Reviewer_8vFk · 2023-10-31

**Soundness:** 2 fair
**Presentation:** 2 fair
**Contribution:** 1 poor
**Rating:** 3
**Confidence:** 4

**Summary:**

This paper proposes to increase the common set of augmentations used in self-supervised learning with a generative model producing new augmentations, which allows to augment the images with new factors of variations. A study on the augmentation parameters using the VICReg method is proposed.

**Strengths:**

1) The idea of using generative models in self-supervised learning is a good idea that has not been explored too much. This paper proposes a first approach and the generated samples shown in Figure 1 look reasonable.

2) The proposed approach is generic to the underlying self-supervised learning method. The authors use VICReg, but any other more recent method could have been used. This general idea could also have application beyond self-supervised learning, anywhere data augmentation is used.

**Weaknesses:**

1) The experimental results are extremely poor. There is no comparison with other methods.

2) The comparison between a setting with and without generative data augmentations is not convincing. The authors choose the VICReg method as the baseline, but the number reported are very low compared to the number reported in the VICReg paper.

3) The gain of using generative augmentation is very limited and might not be worth the burden. An ablation in terms of running time and memory usage compared to the VICReg baseline would be helpful to draw a conclusion. The one proposed in Table 4 is very unclear and doesn’t seem to compare to the baseline.

**Questions:**

Table 2 and 3 are the same table.

---

### Official Review · Reviewer_KP3v · 2023-11-02

**Soundness:** 2 fair
**Presentation:** 2 fair
**Contribution:** 2 fair
**Rating:** 5
**Confidence:** 3

**Summary:**

This paper presents a generative augmentation framework to produce positive views for contrastive learning. Two type of approaches are discussed: noise-based augmentation and mask-based augmentation. The former uses ICGAN and the latter uses MaskGIT.

**Strengths:**

- The paper studies an important problem of constructing views for contrastive learning using generative models.
- The presented results are promising and the experiments are extensive.
- The authors promise to open source the code which will facilitate future research.

**Weaknesses:**

- Clarity: 1) the paper has several long paragraphs of text description that is not super friendly to readers. The presentation might be more clear if formal definitions are given. e.g. when using a GAN generator g, does noise-based augmentation mean g(z+N) where z is the noise vector for image I?
- Clarification: 1) in Table 1, what are the accuracies for baseline (using default augmentations) and what gamma value is used? 2) for mask-based augmentations, do is the ground-truth label used as input to the MaskGIT model?
- It would be interesting to also study other contrastive learning methods like SimCLR. It might also be interesting to discuss other related work e.g. [1-3].
- Problem setting: the mask-based augmentation method utilizes MaskGIT which is trained in a fully-supervised manner and also accepts label as input at inference time. My concern is there might be information leakage during training, which would make the unsupervised or even semi-supervised setting unfair. Also, the method would highly depend on the generative model itself, suppose the generative model is a retrieval model that queries a random image of the same class from training set, then the setting becomes SupCon.

[1] Tamkin, A., Wu, M. and Goodman, N., 2020. Viewmaker networks: Learning views for unsupervised representation learning. arXiv preprint arXiv:2010.07432.
[2] Han, L., Han, S., Sudalairaj, S., Loh, C., Dangovski, R., Deng, F., Agrawal, P., Metaxas, D., Karlinsky, L., Weng, T.W. and Srivastava, A., 2023. Constructive Assimilation: Boosting Contrastive Learning Performance through View Generation Strategies. arXiv preprint arXiv:2304.00601.
[3] Jahanian, A., Puig, X., Tian, Y. and Isola, P., 2021. Generative models as a data source for multiview representation learning. arXiv preprint arXiv:2106.05258.

**Questions:**

please see my questions in the weakness section.